# SARS-CoV-2 ORF3a Induces Incomplete Autophagy via the Unfolded Protein Response

**DOI:** 10.3390/v13122467

**Published:** 2021-12-09

**Authors:** Wen-qing Su, Xue-jie Yu, Chuan-min Zhou

**Affiliations:** State Key Laboratory of Virology, School of Public Health, Wuhan University, Wuhan 430071, China; suwenqing@whu.edu.cn

**Keywords:** coronavirus, SARS-CoV-2, autophagy, ORF3a, UPR

## Abstract

In the past year and a half, SARS-CoV-2 has caused 240 million confirmed cases and 5 million deaths worldwide. Autophagy is a conserved process that either promotes or inhibits viral infections. Although coronaviruses are known to utilize the transport of autophagy-dependent vesicles for the viral life cycle, the underlying autophagy-inducing mechanisms remain largely unexplored. Using several autophagy-deficient cell lines and autophagy inhibitors, we demonstrated that SARS-CoV-2 ORF3a was able to induce incomplete autophagy in a FIP200/Beclin-1-dependent manner. Moreover, ORF3a was involved in the induction of the UPR (unfolded protein response), while the IRE1 and ATF6 pathways, but not the PERK pathway, were responsible for mediating the ORF3a-induced autophagy. These results identify the role of the UPR pathway in the ORF3a-induced classical autophagy process, which may provide us with a better understanding of SARS-CoV-2 and suggest new therapeutic modalities in the treatment of COVID-19.

## 1. Introduction

Autophagy is a conserved cellular process of intracellular degradation of senescent or malfunctioning organelles to maintain intracellular homeostasis [1,2]. Autophagy occurs in response to different forms of stress, including nutrient deprivation, growth factor depletion, infection, and hypoxia. Dysregulated autophagy is associated with certain cancers, neurodegenerative diseases, immune dysfunction, and aging. Thus, autophagy is placed under the spotlight of pharmacologists and clinicians [3]. Autophagy goes through formation of three membrane structures in turn: phagophore, autophagosome, and autolysosome. Briefly, ATGs (autophagy-related genes) are recruited to a specific subcellular location termed the PAS (phagophore assembly site) to form a cup-shaped structure termed the phagophore. Gradually, phagophore elongates and seals into a double-membraned vesicle termed the autophagosome. Autophagosome fuses with the lysosomal membrane to form an autolysosome, which is followed by the degradation of the autophagic body together with its cargo by autolysosomal hydrolytic milieu [2]. ATGs play an irreplaceable role in the whole process of autophagy. To date, more than 30 ATGs have been identified to be involved in the autophagy pathway [4]. Autophagy is modulated by several autophagy-related signaling pathways, including the ULK1 (Unc-51-like kinase 1) complex (consisting of ULK1, ATG13, FIP200, and ATG101) for autophagy initiation, PI3KC3 (the phosphoinositide 3-kinase catalytic subunit type III) complex (consisting of Beclin-1, AMBRA1, VPS34, VPS15, and ATG14L) for autophagosome nucleation, the ATG12-ATG5-ATG16L1 complex for autophagosome elongation, and the STX17-SNAP29-VAMP8 complex for vesicle fusion and further autolysosome-mediated degradation processes [5].

ER (endoplasmic reticulum), a vast membranous network, coordinates diverse cellular processes [6]. To maintain ER homeostasis, cells have evolved multiple protein quality-control systems, including the UPR (unfolded protein response), ERAD (ER-associated degradation), and autophagy. The accumulation of misfolded proteins triggers UPR via three UPR sensors (PERK, IRE1, and ATF6), and the misfolded proteins are cleared by ERAD for proteasomal degradation or autophagy for lysosome-mediated degradation [7,8]. Under basal conditions, PERK, IRE1, and ATF6 are bound by a chaperone binding immunoglobulin protein (BIP) resident on ER membrane. During ER stress, BIP is recruited to unfolded or misfolded proteins and dissociated from these UPR sensors, resulting in UPR activation [9]. It is worth mentioning that the link between autophagy and UPR has been well demonstrated. ATF4 and CHOP, downstream molecules of the PERK-eIF2a pathway, regulate the expression of many ATG genes [10]. IRE1 could mediate Beclin-1 activation through JNK and TRAF2 [11]. ATF6 indirectly regulates autophagy via XBP-1 and CHOP [12].

Coronaviruses are a group of enveloped RNA viruses, with non-segmented, positive-strand RNA genomes, classified under the order *Nidovirales*, family *Coronaviridae*, subfamily *Coronavirinae*. To date, several coronaviruses, including IBV (infectious bronchitis virus), MHV (mouse hepatitis virus), SARS-CoV (Severe Acute Respiratory Syndrome Coronavirus), and MERS-CoV (Middle East Respiratory Syndrome Coronavirus) have been implicated in the induction of autophagy [11,13]. Typically, autophagy is important to facilitate the capture and elimination of invading pathogens [14]. As for coronaviruses, the cellular autophagy pathway is exploited for the replication and egress of coronaviruses under certain situations [15]. Existing evidence supports that SARS-CoV and MHV infection can increase autophagosome formation in host cells. However, it is still controversial whether these viruses rely on autophagy for viral replication and release [16]. COVID-19 is certainly one of the most serious infectious diseases in human history, and people around the world are still trapped in the COVID-19 pandemic due to a lack of comprehensive understanding of the molecular interaction between SARS-CoV-2 and innate immune systems. Of note, structural and nonstructural proteins of coronavirus are essential for interaction with the host innate immune system [17,18], which may shed light on coronavirus invasion and replication. SARS-CoV ORF3a is the largest accessory protein of SARS coronavirus, which is essential for viral replication and release [19,20,21]. SARS-CoV-2 ORF3a and SARS-CoV ORF3a share 72% similarity in the gene sequence, which makes it easy to infer that the two proteins may exhibit similarity in functions.

Investigating the contribution of individual SARS-CoV-2 proteins to innate immunity may allow us to elucidate the pathogenesis of SARS-CoV-2 and speculate effective and specific therapeutics. Here, we focus on the largest accessory protein of SARS-CoV-2 ORF3a and aim to investigate its underlying interaction with autophagy [22,23]. Recently, studies have demonstrated the role of SARS-CoV-2 ORF3a in inducing autophagy in detail [24,25,26]. Here, we seek a more comprehensive mechanism in a different aspect for SARS-CoV-2 ORF3a and autophagy.

## 2. Materials and Methods

### 2.1. Cell Lines and Culture Conditions

Hela, MEF, Vero, Vero-E6, and 293T cells were kept in our laboratory. ATG5 and ATG7 KO MEF cells were kindly provided by Dr. Ming-Zhou Chen (Wuhan University, Wuhan, China). FIP200 and ATG16L1 KO knockout Hela cells were kindly provided by Dr. Feng Shao (Peking University, Beijing, China). Beclin-1 KO Hela cells were kindly provided by Dr. Wen-Sheng Wei (Peking University, China). Cells were cultured in DMEM with 10% heat-inactivated fetal bovine serum and 1% penicillin-streptomycin at 37 °C with 5% CO_2_. Hela, 293T, and MEF cells were usually passaged at 1:4 every other day. Vero and Vero-E6 cells were passaged at 1:6 every other day.

### 2.2. Plasmids and Transfection

SARS-CoV-2 ORF3a (Gene ID: 43740569) with HA-Tag was cloned into the pCAGGS through restriction sites *EcoR*I and *Bam*HI. Plasmids GFP-mCherry-LC3B (#123230) and GFP-LC3B (#11546) were purchased from Addgene (Watertown, MA, USA). Transfection reagent (Yeasen, Shanghai, China) was used according to the manufacturer’s protocol.

### 2.3. Antibodies

Rabbit anti-LC3B (#3868), rabbit anti-ATG5 (#12994), rabbit anti-ATG7 (#8558), rabbit anti-Beclin-1 (#3495), rabbit anti-ATG16L1 (#8089), rabbit anti-BIP (#3177), rabbit anti-ATF6 (#65880), rabbit anti-ATF4(#11815), rabbit anti-XBP1 (#12782), rabbit anti-PERK (#5683), rabbit anti-IRE1 (#3294), rabbit anti-p-eif2α (#3398), rabbit anti-FIP200 (#12436), and mouse anti-CHOP (#2895) antibodies were purchased from Cell Signaling Technology (Danvers, MA, USA) (dilution concentration 1:1000). Rabbit anti-p-IRE1 (PA1-16927) was purchased from Invitrogen (Waltham, MA, USA) (dilution concentration 1:1000). Mouse anti-p62 (18420-1-AP), mouse anti-HA-tag (66006-2-Ig), and mouse anti-β-actin (66009-1-Ig) were purchased from Proteintech (Wuhan, China) (dilution concentration 1:5000). HRP-conjugated goat anti-mouse (G1214) and goat anti-rabbit (G1213) were purchased from Servicebio (Wuhan City, China) (dilution concentration 2:5000). Alexa Fluor 568 goat anti-mouse IgG, IgM (H + L) (A-11004) and Alexa Fluor 647 goat anti-mouse-IgG (H + L) (A-21445) were purchased from Thermo Fisher (Waltham, MA, USA) (dilution concentration 1:500). Immunofluorescence antibodies were diluted in PBS. Immunoblot antibodies were diluted in TBST.

### 2.4. Western Blotting

Cells were lysed in RIPA Lysis Buffer (Beyotime, Shanghai, China) with protease inhibitor cocktail. The total protein concentration was determined using a bicinchoninic acid (BCA) protein assay kit (Beyotime). Equal amounts of total proteins (50 μg) per well were separated with SDS-PAGE (80 V for 30 min, 120 V for 1 h) and transferred to the PVDF membrane (Cytiva, Marlborough, MA, USA) (200 mA for 2 h). Membranes were blocked with 5% non-Fat Milk in TBST for an hour. Cut membranes were incubated with indicated primary antibodies overnight at 4 °C and HRP-labeled secondary antibodies for 2 h. Protein bands were imaged in Amersham Imager 600 system. Photo densitometric data of protein bands were analyzed and quantified with ImageJ.

### 2.5. Confocal Immunofluorescence Microscopy

For immunofluorescence microscopy, 1–2 × 10^4^ cells were plated on a 35 mm Glass Bottom Cell Culture Dish (NEST, Wuxi, China). Treated cells were fixed with 4% paraformaldehyde (Servicebio), permeabilized with 0.02% Triton X-100, and then incubated with indicated antibodies overnight at °C after blocking with 2% BSA. Finally, cells were equilibrated in PBS and stained for DAPI (0.5 μg/mL). Cells were imaged using a Zeiss LSM880292. Microscopy images were possessed with LAS-AF-Lite2.6.0. Representative images of at least three independent replicates are shown.

### 2.6. RNA Isolation and Quantitative Real-Time PCR 

Total RNA was isolated with TRIzol Reagent (Servicebio) according to the instructions and transcribed into the first-strand cDNA with cDNA Synthesis Kit (Servicebio). RT-qPCR (Real-time quantitative PCR) assays were performed using a ChamQ Universal SYBR qPCR Master Mix (Vazyme, Nanjing, China) in a Roche LightCycler 96 system. Data were normalized to the β-actin level. ΔCT(control) = CT(target gene, control) − CT(β-actin, control). ΔCT(treatment) = CT(target gene, treatment) − CT(β-actin, treatment). ΔΔCT = ΔCT(treatment) − ΔCT(control). The primers used in RT-qPCR analysis are listed in Appendix A.

### 2.7. Statistical Analysis

All data were from at least 3 independent trials. Semi-quantitative analysis of Western blotting and immunofluorescence microscopy were conducted with ImageJ. One-way ANOVAs followed by Dunnett’s multiple comparisons test were used in comparisons occurring with 3+ groups. T-tests or T-tests with Welch’s correction were used in experiments where only two groups are compared. All data analyses were performed with GraphPad Prism 8. *p*-values < 0.05 were considered statistically significant (* *p* ≤ 0.05, ** *p* ≤ 0.01, *** *p* ≤ 0.001, **** *p* ≤ 0.001).

## 3. Results

### 3.1. SARS-CoV-2 ORF3a Promoted the Expression of Autophagy-Related Genes

To investigate the interaction between SARS-CoV-2 ORF3a and autophagy, Hela, MEF, and Vero-E6 cells were utilized as cell models to investigate how ORF3a modulated the activation of autophagy. Thus, plasmids pCAGGS and pCAGGS-HA-ORF3a were transfected into Hela, MEF, and Vero-E6 cells. Then we examined the transcription pattern of autophagy-related genes via RT-qPCR (quantitative reverse transcription PCR). Interestingly, our results showed that ORF3a strongly promoted the expression of many autophagy-related genes in Hela, MEF, and Vero-E6 cells, including *ulk1*, *beclin-1*, *wipi1*, *atg5*, *atg7*, and *lc3* (Figure 1 and Appendix A). These data indicate that autophagy could be induced by SARS-CoV-2 ORF3a at least at the transcriptional level.

### 3.2. ORF3a Could Induce Autophagy

To further investigate whether autophagy was induced by SARS-CoV-2 ORF3a, the immunoblotting assay was then used to detect the conversion of LC3B-I to LC3B-II, a standard marker indicating the induction of autophagy. Interestingly, the conversion of LC3B-I to LC3B-II was induced significantly in ORF3a-transfected in Hela, Vero, and MEF cells (Figure 2A,B and Appendix A), indicating that ORF3a-induced autophagy could be a universal phenomenon in these cell lines. To further investigate the interaction between ORF3a and autophagy, autophagy induction was measured over a dose range and over time after transfection with ORF3a. Interestingly, we observed that the induction of autophagy may reach a saturation point when transfected with 1 μg pCAGGS-HA-ORF3a (Figure 2C), while the induction of autophagy by ORF3a was exhibited in a time-dependent manner (Figure 2D). To further identify the role of ORF3a in autophagy induction, pCAGGS-HA-ORF3a was co-transfected with GFP-LC3B into Hela cells. We then used confocal microscopy to measure the distribution of LC3B. Further confocal results showed that LC3B was distributed throughout the cytoplasm in pCAGGS-transfected cells, whereas LC3B was distributed in specific puncta in ORF3a-transfected cells (Figure 2E). These data indicate that autophagy is induced by SARS-CoV-2 ORF3a.

### 3.3. ORF3a-Induced Autophagy Was an Incomplete Process

During autophagy, ubiquitinated proteins are clustered with p62 and are subsequently engulfed by autophagosome. Autophagosome fusing with lysosome results in the degradation of the autophagic body together with its cargo by the autolysosomal hydrolytic milieu [27,28]. The accumulation of autophagy substrate p62 means the degradation process is interrupted. Considering that ORF3a overexpression promoted p62 accumulation in ORF3a-transfected Hela and Vero cells (Figure 2A,B), ORF3a may induce incomplete autophagy. To confirm our hypothesis, mCherry-GFP-LC3B, a tandem fluorescence-labeled LC3B vector, was used to detect autophagic flux based on the switch of fluorescence signals. Briefly, when autophagosome fuses with lysosome, the acidic lysosomal environment could quench the green fluorescence but not the red fluorescence, and red fluorescence can be detected by confocal microscopy assay. Here, ORF3a was co-transfected with mCherry-GFP-LC3B into Hela cells. We observed that ORF3a overexpression could not quench the green fluorescence (Figure 3A), indicating that ORF3a-induced autophagy was incomplete (Appendix A). Rapamycin was employed here as a positive control of autophagy induction. To further confirm the autophagy flux process during ORF3a transfection, Hela cells were subsequently treated with autophagy inhibitors BafA1 (bafilomycin A1) and CQ (chloroquine), which are known to inhibit the autophagosome–lysosome fusion process, to measure the conversion of LC3B-I to LC3B-II and the accumulation of p62 by immunoblotting assay. We observed that pretreatment of BafA1 or CQ could not further promote ORF3a-induced accumulation of LC3B-II and p62 (Figure 3B,C). These data indicate that SARS-CoV-2 ORF3a-induced autophagy is indeed an incomplete process.

### 3.4. ORF3a-Induced Autophagy Was Dependent on the Classical Autophagy 

To determine whether ORF3a-induced autophagy was dependent on classical autophagy processes, a series of autophagy-deficient cells was used, including *Beclin-1* KO HeLa cells and *FIP200* KO HeLa cells. Beclin-1 and FIP200 are classical autophagy-related genes belonging to the complexes that felicitate autophagy initial stage [29,30]. Interestingly, we observed that knockout of *Beclin-1* and *FIP200* in Hela cells partially reversed the ORF3a-induced conversion of LC3B-I to LC3B-II (Figure 4A and Appendix A), implying that ORF3a-induced autophagy was classic and dependent on Beclin-1/FIP200. To further verify ORF3a-induced classic autophagy, conversion of LC3B-I to LC3B-II was detected in *Atg5* or *Atg7* KO MEF cells and *Atg16L1* KO Hela cells. Consistently, Atg16*L1* KO HeLa cells and *Atg7* KO MEF cells abolished ORF3a-induced conversion of LC3B-I to LC3B-II (Appendix A). Intriguingly, ORF3a could induce a small amount of LC3B-II in *Atg5* KO MEF cells, suggesting that ORF3a-induced autophagy may not be completely dependent on Atg5. Moreover, two autophagy initial stage inhibitors, PI(3)K inhibitor 3-MA (3-methyladenine) and Ca^2+^ chelator BAPTA-AM, were further employed to assess the ORF3a-induced autophagy. Further immunoblotting results showed that ORF3a-induced autophagy was inhibited by 3-MA (Figure 4B) and BAPTA-AM (Figure 4C). These results indicate that ORF3a-induced autophagy is dependent on the classical autophagy process.

### 3.5. ORF3a Induced Autophagy through Unfolded Protein Response

Like autophagy, UPR (the unfolded protein response) is a conserved cellular stress response induced by ER (endoplasmic reticulum) stress that is important for maintaining cell homeostasis in response to various pathogenic infections [31]. Currently, many studies suggest that autophagy is tightly regulated by the UPR. The downstream molecules of UPR can directly induce the transcription of autophagy-related genes; besides, UPR is also able to trigger autophagy directly by affecting Beclin-1 [11]. In mammalian cells, there are three unique UPR signal pathways: PERK-ATF4, ATF6, and IRE1-XBP1. BIP is a chaperone and master regulator of the UPR. To elucidate how ORF3a induced autophagy, immunoblotting assay was used to detect UPR-related proteins BIP, ATF4, CHOP, XBP1-s, and ATF6 to investigate whether ORF3a could induce the UPR pathway. Interestingly, we found that UPR-related proteins BIP, ATF4, CHOP, ATF6-C (50 kDa), and XBP1-s were increased and ATF6 (90 kDa) was decreased in ORF3a-transfected Hela cells (Figure 5A), indicating the induction of three UPR signal pathways. To determine which UPR pathway was involved in ORF3a-induced autophagy process, three UPR inhibitors (Ceapin-A7 for ATF6, GSK2606414 for PERK-ATF4, and 4μ8C for IRE1-XBP1) were used. We found that ORF3a-induced conversion of LC3B-I to LC3B-II was largely impaired by Ceapin-A7 and 4μ8C, while not GSK2606414 (Figure 5B and Appendix A). To further solidify our UPR inhibitor results, siRNAs were used to knock down the expression of *atf6*, *perk*, and *ire1* in Hela cells. We observed that transfection of ATF6 siRNA or IRE1 siRNA inhibited the ORF3a-induced conversion of LC3B-I to LC3B-II, while transfection of PERK siRNA did not (Figure 5C–E and Appendix A). These results indicate that ORF3a promotes the induction of autophagy via classic ATF6 and IRE1-XBP1 UPR pathway.

## 4. Discussion

To date, many studies have demonstrated the important role of SARS-CoV ORF3a in interacting with intracellular immune responses, including apoptosis [32], ER stress [33], NLRP3 inflammasome [34], and type I interferon signaling pathway [33]. Importantly, knowledge of virus–host interaction is essential to develop effective acting antiviral therapies, such as drugs targeting the autophagy process [35]. Herein, we focus on the interaction between SARS-CoV-2 ORF3a and autophagy.

In our study, we found that SARS-CoV-2 ORF3a induced the conversion of LC3B-I to LC3B-II and p62 accumulation, suggesting that ORF3a might induce incomplete autophagy. When evaluating the cellular autophagy, it is necessary to detect the changes of autophagy flux, dynamic changes of autophagosome formation, fusion of autophagosome and lysosome, substrate degradation, and so on. This conclusion was further confirmed by using the autophagy flux reporter plasmid mCherry-GFP-LC3B and the autophagy flux inhibitors Baf-A1 and CQ [36]. Consistently, the latest study indicates that SARS-CoV-2 ORF3a interacts with VPS39 interaction to prevent the assembly of the STX17-SNAP29-VAMP8 SNARE complex in forming autolysosome [25]. Moreover, the cytoprotective properties of autophagy have raised the particular interest of scientists and clinicians. Antimalarial drugs CQ and HCQ (hydroxychloroquine) exert strong anti-SARS-CoV-2 effects in SARS-CoV-2 infected Vero E6 cells but fail to inhibit SARS-CoV-2 replication in TMPRSS2-expressing human lung cells or animal models and COVID-19 patients [37,38]. It is doubtful that their putative effects on autophagy inhibition are necessarily causal for their anti-SARS-CoV-2 activity. Detailed mechanisms of their action against SARS-CoV-2 infection and replication still need to be elucidated. Combined with all current studies on SARS-CoV-2 ORF3a, ORF3a may promote viral replication to some extent by promoting the formation of autophagosomes. On the other hand, ORF3a inhibits the formation of autolysosomes to protect the virus from the hydrolysis in acidic environments.

To further understand ORF3a-induced autophagy, autophagy-related gene deficient cells were used to detect classical autophagy. We found that autophagy induced by ORF3a was decreased in FIP200 and Beclin-1 deficient cells. It is known that FIP200 belongs to Atg1/ULK complex which is a stable complex responsible for autophagy initiation. In mammals, Beclin-1 is an essential component of two PI3K complexes through its interaction with either Barkor (Beclin-1-associated autophagy-related key regulator)/ATG14 complex or UVRAG (UV radiation resistance-associated gene) [39]. It is reported that SARS-CoV-2 ORF3a interacts with autophagy regulator UVRAG to inhibit Beclin-1-Vps34-UVRAG complex, which results in the accumulation of formed autophagosomes in HeLa cells [24,26]. To further confirm that ORF3a could induce classic autophagy, we used another batch of ATG-deficient cells. ATG7 and Atg16L1 deficient cells abolished ORF3a-induced LC3B-II accumulation. ATG5, ATG7, and Atg16L1 belong to the Atg12-ATG5-Atg16(L) dimer which is important for Atg8/LC3-PE conjugation [40]. Interestingly, ORF3a still induced a small amount of LC3B-II in ATG5-deficient MEF cells, indicating that ORF3a may bypass ATG5 to some extent to cause autophagy. 3-MA is known to permanently inhibit autophagosome formation by PI3K class I and transiently inhibit PI3K class III. BATPA-AM is a Ca^2+^ chelator that can penetrate the cell membrane to disrupt intracellular signal transduction including mTORC1 and Beclin-1 to inhibit autophagy in this case [41]. ORF3a-induced autophagy in Hela cells was impaired by 3-MA or BATPA-AM. Combined with autophagy inhibitors 3-MA and BATPA-AM, these results indicated that FIP200-Beclin-1-mediated classical autophagy pathway was essential for ORF3a-induced autophagy. This could be a supplement to the detailed study of ORF3a-induced autophagy in the previous studies [24,26]. Besides, researchers find that Beclin-1 stabilizing drug niclosamide could suppress SARS-CoV-2 replication through autophagy activation in SARS-CoV-2-infected Vero-FM cells [42], which may give us some hints to combat the virus through classic autophagy. The underlying induction mechanism is worth further investigation.

As a part of cellular stress, autophagy is tightly connected with ER [43]. SARS-CoV ORF3a, ORF6, ORF8, and spike (S) protein are reported to induce ER stress [33,44,45,46]. UPR is known as part of the cellular ER stress for coping with protein-folding alterations [47], which comprises PERK-ATF4, ATF6, and IRE1-XBP1—three downstream pathways [48]. Non-structural proteins such as Nsp3, 4, and 6 of SARS-CoV are located on the ER membrane and proved to promote double-membrane vesicle formation that could be associated with viral replication and transcription complex [49]. To further explore the relationship between autophagy and UPR under SARS-CoV-2 ORF3a transfection, three classical UPR inhibitors and siRNAs were used [50,51,52]. We found that ATF6 and IRE1-XBP1 deficiency inhibited the ORF3a-induced conversion of LC3B-I to LC3B-II, while UPR deficiency could also block the accumulation of LC3B-II induced by CQ (Appendix A). We cannot deny the fact that UPR deficiency could block basal autophagy, whereas our results confirmed that ORF3a can indeed promote the activation of UPR, supporting that ORF3a-induced autophagy is dependent on the ATF6 and IRE1 signal pathway. In turn, ORF3a-induced UPR was abolished in Beclin-1 KO Hela cells (Appendix A). Actually, the role of autophagy in UPR remains to be investigated, and they could take on functions in a context-dependent manner. Earlier studies indicate that the ATG5-ATG7-ATG16L1 complex can relieve ER-stress and impair the activation of UPR [53,54]. In addition, Beclin-1, belonging to R-BIP/Beclin-1/p62 complex, may promote the activation of UPR [55]. These results indicate the complex interaction between UPR and autophagy during ORF3a transfection, which remains for further investigation.

In conclusion, our results not only identified the important role of the classical autophagy genes FIP200 and Beclin-1 in the ORF3a-induced incomplete autophagy process, but also elucidated the crosstalk between UPR and classical autophagy under SARS-CoV-2 ORF3a transfection status. It should be important to expand our understanding of the relationship between SARS-CoV-2 and autophagy. This may have directional significance for the new therapeutic modalities in the treatment of COVID-19 and allow us to further investigate the effects of UPR on FIP200-Beclin-1 axis activation and viral replication to combat the virus. Limitations of our study also exist. It is elusive whether ORF3a-induced autophagy is dependent on the transcription of autophagy-related genes. It remains unclear whether an additional pathway is involved in ORF3a-induced autophagy, and how ORF3a promotes the activation of FIP200 and Beclin-1. 

## Figures and Tables

**Figure 1 viruses-13-02467-f001:**
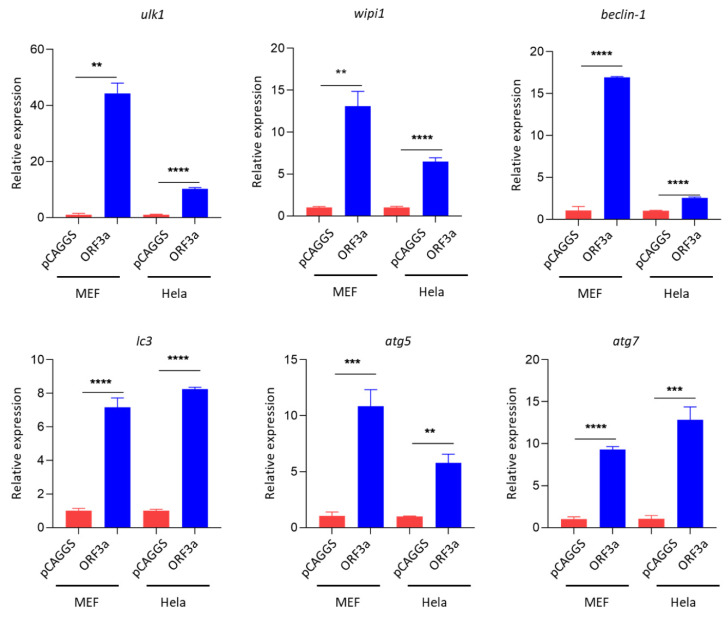
SARS-CoV-2 ORF3a promoted the expression of autophagy-related genes. Hela and MEF cells (1–2 × 10^5^ cells) were transfected with HA-Tag-ORF3a (2 μg) or empty vector (2 μg) for 24 h. qPCR was performed to detect the expression level of *ulk1*, *atg13*, *beclin-1*, *wipi1*, *atg5*, *atg7*, *p62,* and *lc3*. Data were analyzed by T-tests or T-tests with Welch’s correction. qPCR Data (mean ± SEM) are representative of three independent experiments. ** *p* ≤ 0.01, *** *p* ≤ 0.001, **** *p* ≤ 0.001.

**Figure 2 viruses-13-02467-f002:**
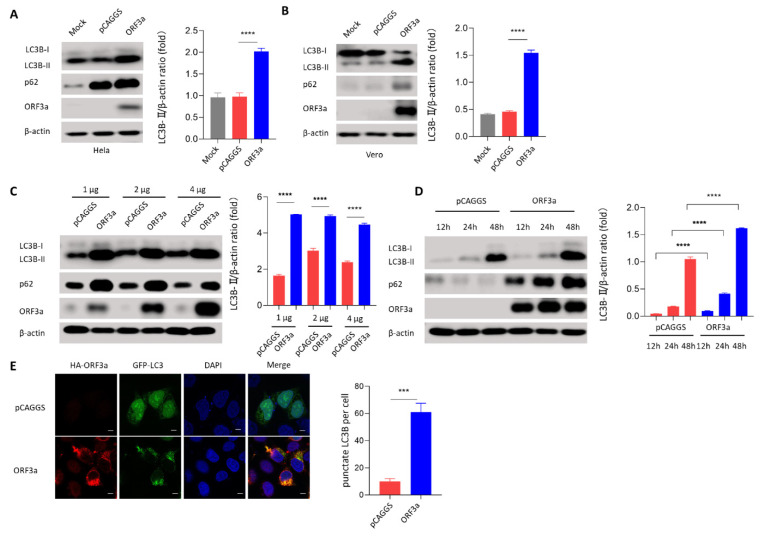
ORF3a could induce autophagy. (**A**,**B**) Hela and Vero cells (1–2 × 10^5^ cells) were transfected with HA-Tag-ORF3a (2 μg) or empty vector (2 μg) for 24 h. Western blot was performed to detect LC3B and p62. (**C**) Hela cells (1–2 × 10^5^ cells) were transfected with HA-Tag-ORF3a (1, 2, and 4 μg) or empty vector (1, 2, and 4 μg) for 24 h. Western blot was performed to detect LC3B and p62. (**D**) Hela cells (1–2 × 10^5^ cells) were transfected with HA-Tag-ORF3a (2 μg) or empty vector (2 μg) for 12 h, 24 h, and 48 h. Western blot was performed to detect LC3B and p62. (**E**) Hela cells (1–2 × 10^4^ cells) were co-transfected with GFP-LC3B (0.5 μg) and HA-Tag-ORF3a (0.5 μg) or empty vector (0.5 μg) for 24 h. Immunofluorescence was performed to detect LC3B puncta. Scale bars: 4 μm. Magnification: 630×. One-way ANOVAs followed by Dunnett’s multiple comparisons test were used in comparing multiple groups. Student’s t-test was used to test statistical significance across two groups. Data were representative of 3 independent experiments. *** *p* ≤ 0.001, **** *p* ≤ 0.001.

**Figure 3 viruses-13-02467-f003:**
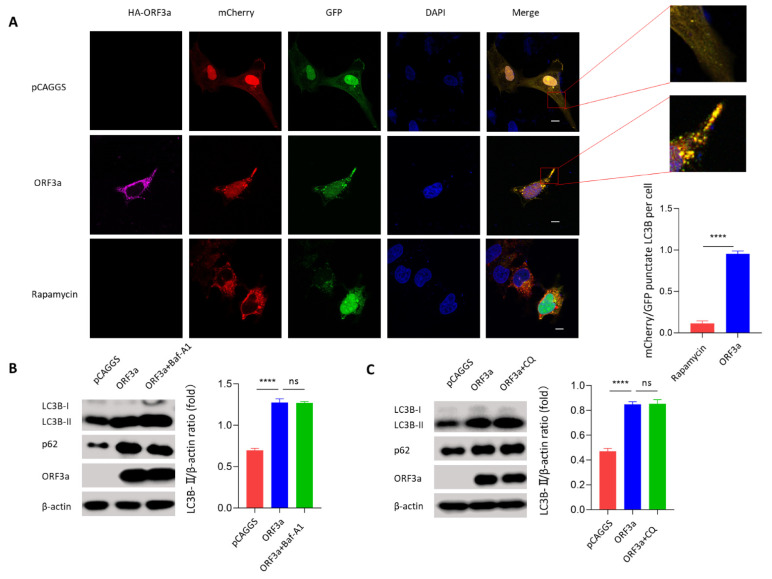
**ORF3a-induced autophagy was an incomplete process.** (**A**) Hela cells (1–2 × 10^4^ cells) were co-transfected with mCherry-GFP-LC3B (1 μg) and HA-Tag-ORF3a (1 μg) or empty (1 μg) vector for 24 h. Rapamycin (50 mM, 6 h) was used as a positive control. Partial magnifications of 10× based on the original images were placed next to the original ones. Immunofluorescence was performed to assess autophagy status. Scale bars: 4 μm. Magnification: 630×. (**B**) Hela cells (1–2 × 10^5^ cells) were transfected with HA-Tag-ORF3a (2 μg) for 24 h and treated with Baf-A1 (50 nM) for 6 h before harvest. Western blot was performed to detect LC3B and p62. (**C**) Hela cells (1–2 × 10^5^ cells) were transfected with HA-Tag-ORF3a (2 μg) for 24 h and treated with CQ (50 nM) for 6 h before harvest. Western blot was performed to detect LC3B and p62. One-way ANOVAs followed by Dunnett’s multiple comparisons test were used in comparing multiple groups. Student’s *t*-test was used to test statistical significance across two groups. Data were representative of 3 independent experiments. **** *p* ≤ 0.001.

**Figure 4 viruses-13-02467-f004:**
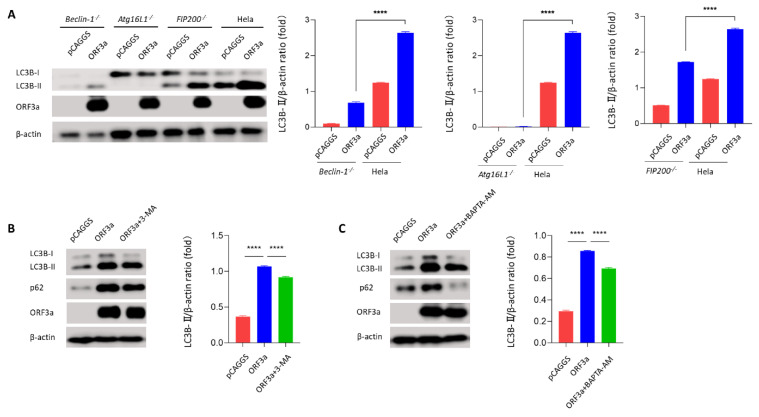
**ORF3a-induced autophagy was dependent on the classical autophagy.** (**A**) *Beclin-1* KO, *Atg16L1* KO, *FIP200* KO, and WT Hela cells (1–2 × 10^5^ cells) were transfected with HA-Tag-ORF3a (2 μg) or empty vector (2 μg) for 24 h. Western blot was performed to detect LC3B. (**B**) Hela cells (1–2 × 10^5^ cells) were transfected with HA-Tag-ORF3a (2 μg) for 24 h and treated with 3-MA (50 μM) for 6 h before harvest. Western blot was performed to detect LC3B and p62. (**C**) Hela cells (1–2 × 10^5^ cells) were transfected with HA-Tag-ORF3a (2 μg) for 24 h and pretreated with BATPA-AM (50 nM) for 16 h before harvest. Western blot was performed to detect LC3B and p62. One-way ANOVAs followed by Dunnett’s multiple comparisons test were used in comparing multiple groups. Data were representative of 3 independent experiments. **** *p* ≤ 0.001.

**Figure 5 viruses-13-02467-f005:**
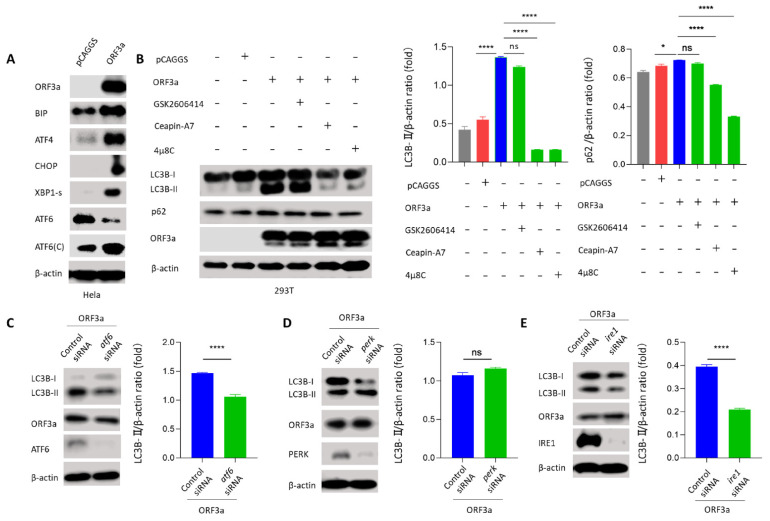
**ORF3a induced autophagy through unfolded protein response.** (**A**) Hela cells (1–2 × 10^5^ cells) were transfected with HA-Tag-ORF3a (2 μg) or empty vector (2 μg) for 24 h. Western blot was performed to detect UPR-related proteins. (**B**) 293T cells (1–2 × 10^5^ cells) were transfected with HA-Tag-ORF3a (2 μg) or empty vector (2 μg) for 24 h pretreated with Ceapin-A7 (10 nM), GSK2606414 (5 nM), 4μ8C (10 nM) for 16 h before harvest. Western blot was performed to detect LC3B and p62. (**C**–**E**) Hela cells (1–2 × 10^5^ cells) were transfected with HA-Tag-ORF3a (2 μg) and *atf6*/*perk*/*ire1* siRNA (50 nM) or control siRNA (50 nM) for 36 h. Western blot was performed to detect LC3B. One-way ANOVAs followed by Dunnett’s multiple comparisons test were used in comparing multiple groups. Student’s t-test was used to test statistical significance across two groups. Data are representative of 3 independent experiments. * *p* ≤ 0.05, **** *p* ≤ 0.001.

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
