# Peer review of "SARS-CoV-2 ORF3a Induces Incomplete Autophagy via the Unfolded Protein Response"

_viruses, 2021, doi:10.3390/v13122467_

Round 1

Reviewer 1 Report

The manuscript entitled “SARS-COV2 ORF3a induces incomplete autophagy via the unfolded protein response” has been reviewed. The authors reported the crosstalk between UPR and classical autophagy under SARS-CoV-2 ORF3a transfection status. Obtained results lead to better understanding of SARS- CoV-2 and suggest new therapeutic modalities in the treatment of COVID-19. Data were collected and evaluated in an appropriate manner and the paper is well written. There are some minor comments as follows:

- It is better to include some explanation about the UPR pathway in the introduction.

- Figure1, Figure S1, Figure S2 and Figure S3 have just one part. Therefore, it is not necessary to include "Part A" in these figures and their legends.

- Page 4, Line 170, promotion of p62 accumulation in ORF3a-transfected Hela and Vero cells is not shown in Figure S2A. It should be omitted in the parentheses.

Author Response

  1. It is better to include some explanation about the UPR pathway in the introduction.

Response: Thank you for your wonderful comments. We have added relative description of UPR in introduction part (line 46-58).

  1. Figure1, Figure S1, Figure S2 and Figure S3 have just one part. Therefore, it is not necessary to include "Part A" in these figures and their legends.

Response: We have deleted the “part A” and “A” in relative section.

  1. Page 4, Line 170, promotion of p62 accumulation in ORF3a-transfected Hela and Vero cells is not shown in Figure S2A. It should be omitted in the parentheses.

Response: We have adjusted the description as suggested (line 182).

Reviewer 2 Report

Using several approaches, currently used to investigate the autophagic modulation, the authors shows that Hela, Vero or MEF cells overexpressing OFR-3 protein from SAR-Cov2 induces an accumulation of autophagosome in cells. This autophagosome accumulation is dependent of classical autophagic genes involved in autophagy initiation and seem to be dependent of some UPR response (IRE and ATF6 but not PERK pathways). While the experiment is well performed and results are convincing, this work will not bring new information to understand the relation between SAR-Cov2 and autophagy. Moreover, some conclusions are not convincing and require more experimentations to be demonstrate.

Major comment:

  1. In this very competitive field, many publications have already demonstrated a clear relation between SAR-Cov2-ORF3a and autophagy, showing that this viral protein inhibit the autophagic maturation step by directly targeting the fusion and the function of autolysosome (e.g., Miao,G et al, Dev cell 2021, Qu, y et al, bioRxiv, 2020, Zhanb, g et al, Cell Discov 2021…). The authors claim that their work bring new information by saying that “ORF3a-induced autophagy” required the classical level of autophagy and some UPR responses. However, as explained in my others comments below, these conclusions are not convincing.
  2. The authors should carefully use the sentence “ORF3a-induced autophagy”. It’s not clear here if expression of ORFa really induce de novo autophagosome. Regarding literature and authors experiments, ORF3a expression did not induce autophagy but block basal autophagy. Indeed, when authors used BAF or CQ to block autophagosomes fusion, their did not find differences for LC3-II conversion between ORF3 and ORFa-BAF or ORF3a-CQ cells, demonstrating that there are not induction of de novo autophagosome in the presence of ORF3a. Based on this observation, the fact that LC3-II conversion was less observed in Beclin-1, FIP200 or Atg16L1 KO cells after the transfection of ORF3a is probably due to the lower expression of basal autophagy in these cells, and is not specific to ORF3a-induced autophagosomes. The authors should try to reiterate these experiments using BAF or CQ as the control (instead of ORF3a expression). In my opinion, similar results should be observed with these inhibitors.
  3. Similar observation should be done for the UPR implication. Again, the role of UPR responses in the ORF3a-indcued autophagosome in not clear. Inhibition of UPR response could decrease the basal autophagy, and the decrease of LC3-II conversion observed in ORF3-expressing cells after UPR inhibition may be due to indirect effect of this drug on basal autophagy and may not be specific to ORF3a expression. The authors should try to use inhibitor or siRNA for UPR response in cells treated with CQ or BAF (instead of ORF3a expression) to see if a similar decrease of autophagy should be overserved. Moreover, several papers demonstrated that impairment of autophagy (including the maturation step of autophagy) can induce ER and/or UPR responses. It’s possible that the increase of UPR responses observed in ORF3a expressing cells may be due to the inhibition of autophagic maturation step. Again, the authors should try to block autophagy with inhibitors of maturation to demonstrated that UPR responses is not induced by the blockage of autophagy mediated by ORF3a.

Author Response

  1. The authors should carefully use the sentence “ORF3a-induced autophagy”. It’s not clear here if expression of ORF3a really induce de novo autophagosome. FIP200 or Atg16L1 KO cells after the transfection of ORF3a is probably due to the lower expression of basal autophagy in these cells, and is not specific to ORF3a-induced autophagosomes. The authors should try to reiterate these experiments using BAF or CQ as the control (instead of ORF3a expression).

Response: Thanks for your very critical suggestions. As suggested, we detected the protein level of LC3B in Beclin-1 and FIP200 KO Hela cells. CQ was used as the positive control. We observed that CQ-induced LC3B-II accumulation was impaired in Beclin-1 and FIP200 KO Hela cells (Figure S4D, E). Importantly, although previous studies revealed that ORF3a can block the fusion of autophagosome and lysosome [1, 2], the role of ORF3a in the autophagy induction was also introduced, which interacts with UVRAG and thus promote Beclin-1 dependent autophagosome formation [3]. Combined with our qPCR-results on detecting the expression of autophagy-related genes and our immunoblotting results by using different KO cells or autophagy inhibitors, our study supported that ORF3a can promote the induction of autophagy, which could be mediated via UPR or other underlying signal pathways. Relevant discussion was also added in the manuscript (line 347-350).

Reference:

[1] Miao, G.; Zhao, H.; Li, Y.; Ji, M.; Chen, Y.; Shi, Y.; Bi, Y.; Wang, P.; Zhang, H., ORF3a of the COVID-19 virus SARS-CoV-2 blocks HOPS complex-mediated assembly of the SNARE complex required for autolysosome formation. Dev Cell 2021, 56, (4), 427-442 e5.

[2] Zhang, Y.; Sun, H.; Pei, R.; Mao, B.; Zhao, Z.; Li, H.; Lin, Y.; Lu, K., The SARS-CoV-2 protein ORF3a inhibits fusion of autophagosomes with lysosomes. Cell Discov 2021, 7, (1), 31.

[3] Qu, Y. F.; Wang, X.; Zhu, Y. K.; Wang, W. L.; Wang, Y. Y.; Hu, G. W.; Liu, C. R.; Li, J. J.; Ren, S. H.; Xiao, M. Z. X.; Liu, Z. S.; Wang, C. X.; Fu, J.; Zhang, Y. C.; Li, P.; Zhang, R.; Liang, Q. M., ORF3a-Mediated Incomplete Autophagy Facilitates Severe Acute Respiratory Syndrome Coronavirus-2 Replication. Front Cell Dev Biol 2021, 9

  1. The authors should try to use inhibitor or siRNA for UPR response in cells treated with CQ or BAF (instead of ORF3a expression) to see if a similar decrease of autophagy should be overserved.

Response: Hela cells were pretreated with siRNAs to knockdown the expression of UPR-related ATF6, PERK, or IRE1, and then treated with CQ. Our results indicated that ATF6, IRE1 or PERK deficiency significantly blocked the protein level of LC3B-II induced by CQ (Figure S5C-E). In addition, we found that ATF6 and IRE1-XBP1 deficiency inhibited the ORF3a-induced conversion of LC3B-I to LC3B-II, while PERK not (Figure 5). We cannot deny the fact that UPR deficiency could block basal autophagy, whereas our results confirmed that ORF3a can indeed promote the activation of UPR, supporting that ORF3a-induced autophagy is dependent on ATF6 and IRE1 signal pathway. Relevant discussion was also added in the manuscript (line 384-388).

  1. the authors should try to block autophagy with inhibitors of maturation to demonstrated that UPR responses is not induced by the blockage of autophagy mediated by ORF3a.

Response: As suggested, we detected the activation of UPR in Beclin-1 KO Hela cells. We found that ORF3a-induced UPR was impaired in Beclin-1 KO Hela cells (Figure S5F). However, the feedback from autophagy to the UPR signaling pathways has not been implicated clear, thus they could be functioning in a context-dependent manner. Earlier studies indicate that ATG5-ATG7-ATG16L1 complex can relief ER-stress and impair the activation of UPR [1,2]. In addition, Beclin-1 belongs to R-BIP/Beclin-1/p62 complex may promote the activation of UPR [3]. These results indicate the complex interaction between UPR and autophagy during ORF3a transfection, which remains for further investigation (line 388-394).

Reference:

[1] Adolph, T. E.; Tomczak, M. F.; Niederreiter, L.; Ko, H. J.; Bock, J.; Martinez-Naves, E.; Glickman, J. N.; Tschurtschenthaler, M.; Hartwig, J.; Hosomi, S.; Flak, M. B.; Cusick, J. L.; Kohno, K.; Iwawaki, T.; Billmann-Born, S.; Raine, T.; Bharti, R.; Lucius, R.; Kweon, M. N.; Marciniak, S. J.; Choi, A.; Hagen, S. J.; Schreiber, S.; Rosenstiel, P.; Kaser, A.; Blumberg, R. S., Paneth cells as a site of origin for intestinal inflammation. Nature 2013, 503, (7475), 272-+.

[2] Zheng, W.; Xie, W. W.; Yin, D. Y.; Luo, R.; Liu, M.; Guo, F. J., ATG5 and ATG7 induced autophagy interplays with UPR via PERK signaling. Cell Communication and Signaling 2019, 17.

[3] Song, X. X.; Lee, D. H.; Dilly, A. K.; Lee, Y. S.; Choudry, H. A.; Kwon, Y. T.; Bartlett, D. L.; Lee, Y. J., Crosstalk Between Apoptosis and Autophagy Is Regulated by the Arginylated BiP/Beclin-1/p62 Complex. Mol Cancer Res 2018, 16, (7), 1077-1091.

Round 2

Reviewer 2 Report

I thanks the authors for their responses and additions (text and experiments) in the manuscript. 

While the role of ORF3a in the induction of new autophagosome is not really clear her, we can observed that CQ treatment is regulated by all (three) UPR pathways and ORF3a-modulated autophagy is only mediated by two, which suggest that ORF3a could modulated autophagy in a specific manner, and not by just blocking the basal autophagy. 

Manuscript has been improved enough for publication.